# When it Rains, it Pours: Modeling Media Storms and the News Ecosystem

**Benjamin Litterer**
School of Information
University of Michigan
blitt@umich.edu

**David Jurgens**
School of Information and
Computer Science & Engineering
University of Michigan
jurgens@umich.edu

**Dallas Card**
School of Information
University of Michigan
dalc@umich.edu

## Abstract

Most events in the world receive at most brief coverage by the news media. Occasionally, however, an event will trigger a *media storm*, with voluminous and widespread coverage lasting for weeks instead of days. In this work, we develop and apply a pairwise article similarity model, allowing us to identify story clusters in corpora covering local and national online news, and thereby create a comprehensive corpus of media storms over a nearly two year period. Using this corpus, we investigate media storms at a new level of granularity, allowing us to validate claims about storm evolution and topical distribution, and provide empirical support for previously hypothesized patterns of influence of storms on media coverage and intermedia agenda setting.

## 1 Introduction

On March 9, 2021 jury selection began in the trial of Minneapolis police officer Derek Chauvin. Over the next month and a half, details of the case such as calls for delay, jury selection, presentation of evidence, and an eventual guilty verdict were covered scrupulously by the media—peaking in coverage 48 days after the story first broke (see Figure 1). This type of enduring and high volume coverage is a prime example of a *media storm*. In this paper, we construct a comprehensive corpus of media storms from nearly two years of coverage, which we use to study the nature and dynamics of this particularly influential media event type.

Citizens are exposed to major news stories such as these on a daily basis, and these news events have been shown to influence public discussion and debate (Walgrave et al., 2017; McCombs and Shaw, 1972; King et al., 2017; Langer and Gruber, 2021). Furthermore, decades of scholarship have attempted to characterize this type of coverage under a range of theoretical definitions such as media hypes, political waves, and news waves (Wien and

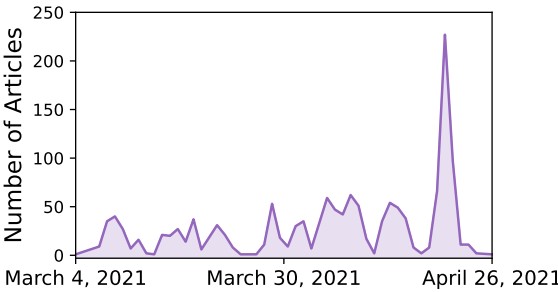

Figure 1: The evolution of the media storm surrounding Derek Chauvin's trial, shown as the number of articles published about this story per day. Coverage peaked on April 20th, the day a guilty verdict was announced.

Elmelund-Præstekær, 2009; Wolfsfeld and Sheafer, 2006; Kepplinger and Habermeier, 1995). We build especially on the work of Boydstun et al. (2014), who approach this topic by synthesizing previous theoretical definitions under the term media storms, focusing on coverage so prominent that media consumers can't help but know about it. While prior research has theorized about their mechanisms, there has been limited quantitative analysis of the conditions in which storms emerge, and the effects that they have on the news ecosystem.

Through the creation of a state-of-the-art news article similarity model, we identify story clusters from a corpus of over 4 million news articles. Within this larger dataset, we identify 98 media storms over a period of 21 months; while small in number, these storms alone comprise 20,331 articles in our data, and were covered widely in 417 mainstream, conspiratorial, political, and local news outlets. The diversity and magnitude of our media storms provide a unique opportunity to test specific hypothesis from Media and Communications theory.

In brief, this work makes the following contributions: We create a state-of-the-art article embedding model enabling fast and accurate comparison of billions of news article pairs, and use it to create

and publicly release a new dataset of story clusters, augmenting a combination of existing news datasets representing almost two years of coverage (§3). Working at a more fine grained level of analysis than was previously possible, we then validate and strengthen previous observations about the temporal dynamics (§4) and topical composition (§5) of media storms. Finally, we provide evidence about the role of media storms in regard to intermedia agenda setting and gatekeeping patterns amongst news outlets (§6).

Given the influence of media storms and other news on democratic debate and public discourse (Walgrave et al., 2017; King et al., 2017), our findings also offer a step towards answering the question of which actors get to decide what information becomes impossible to ignore. Furthermore, the introduction of our model allows researchers to trace the evolution of single stories and events over time, enabling a number of new downstream analyses in the agenda setting and framing space. Our model for predicting article similarity, as well as data and replication code for this paper, are available at `https://github.com/blitt2018/mediaStorms`.

## 2 Background

**Defining Media Storms** A number of definitions have been employed to characterize explosive and long-lasting stories that arise in the news ecosystem (Vasterman, 2018). Kepplinger and Habermeier (1995) present the idea of news waves, which are started by a key event and trigger increased coverage of similar events. Vasterman (2005) builds off of the news wave phenomenon, introducing media hypes. Whereas news waves are identified based on an increase in coverage relative to the number of events, media hypes are defined by a self-reinforcing process of news production resulting in an increased number of articles pertaining to a single issue. Boydstun et al. (2014)'s media storm abstracts the descriptive claims of prior theories while relaxing assumptions about the mechanisms behind them. Theoretically, media storms are when the media ecosystem goes into "storm mode", shifting from routine coverage to an intensive focus on a particular event, issue, or topic over a period of time (Hardy, 2018).

**Mechanisms** Two primary mechanisms have been identified to explain the development of media storms. First, media storms may be caused by a lowering of the gatekeeping threshold (Kep-

plinger and Habermeier, 1995; Brosius and Eps, 1995; Hardy, 2018; Wolfsfeld and Sheafer, 2006). After an event or issue has received extensive coverage at a given outlet, it is seen as more newsworthy by reporters and editors at that same outlet, who will therefore be more likely to write or publish additional coverage of it. This notion is supported by evidence that organizations tend to operate with a centralized, top down structure in the wake of large events, which implies that certain trigger events cause a large spike in follow up coverage and a subsequent media storm (Hardy, 2018; Wien and Elmelund-Præstekær, 2009).

While the gatekeeping threshold primarily concerns within-outlet reporting practices, the intermedia agenda setting hypothesis describes the emulation of coverage between outlets. When outlets see that a story has risen to prominence in other outlets, they may be incentivized to shift their own coverage (Boydstun et al., 2014; Hardy, 2018; Vliegenthart and Walgrave, 2008; Anderson, 2010; Vargo and Guo, 2017). This incentivization may be especially true for outlets which feel that their principal reporting domains are being encroached upon (Boydstun et al., 2014).

## 3 Creating a Media Storms Corpus

To investigate media storms at scale, we augment a set of three large and diverse datasets by creating a model to group news articles into story clusters.

### 3.1 Data

To capture coverage dynamics from a wide variety of sources, we use the NELA-GT-2020, NELA-GT-2021, and NELA-Local datasets (Gruppi et al., 2021; Horne et al., 2022). NELA-GT-2020 is composed of nearly 1.8 million articles published over the course of 2020. Notably, this coverage from 519 sources varies widely in focus, including political, conspiratorial, and health-oriented content alongside coverage from mainstream outlets such as The New York Times and The Washington Post.

NELA-GT-2021 has a similar outlet distribution and contains over 1.8 million articles from 367 outlets in 2021. In addition, this data includes Media Bias/Fact Check ground truth scores for reliable, mixed, and unreliable coverage. We also included local news from the NELA-Local database, which consists of more than 1.4 million articles from 313 outlets in 46 states. NELA-Local only spans from April 1, 2020 – December 31, 2021, so NELA-

GT 2020 was truncated to match this range. In all NELA datasets, news was collected from RSS feeds, meaning that our study focuses on online, rather than print media. After merging the three datasets and removing duplicate articles from the same outlet, our data consists of 4.2 million articles from 815 unique outlets.

## 3.2 News Similarity Model

In order to group news articles together into story clusters, we created a model to identify whether two articles are about the same story. Here, we make use of the dataset from the SemEval task on news article similarity (Chen et al., 2022), which contains labels for 4,950 pairs of news articles rated for similarity.[1] Our goal was to create an accurate model that could scale to millions of articles. To do so, we fit a bi-encoder MPNet model initialized with `all-mpnet-base-v2` parameters (Song et al., 2020).[2] We fine-tuned this model by minimizing mean squared error between the cosine similarity of article pairs embedded by the model and the corresponding human similarity ratings.

Our approach incorporates multiple techniques from models submitted to the SemEval competition (Chen et al., 2022). To ensure scalability, we used a bi-encoder approach; at interference time, this allows us to precompute all $n$ article embeddings ($O(n)$), rather than using a cross-encoder, (as was used by the top performing model in the competition, Xu et al., 2022), which requires $O(n^2)$ to encode and evaluate all pairs. The model from Wangsadirdja et al. (2022) also used a bi-encoder approach but we simplified and streamlined their method for better scalability by discard the additional convolutional layers and the ensembling strategy used by these authors.

Inspired by Xu et al. (2022), we concatenate the first 288 tokens of each article with the last 96 tokens, in order to include information from both the head and tail of each article. Of the 4,950 article pairs available, 1,791 were English-English pairs and the remaining 3,159 were non-English. In order to augment this data, (as done by Xu et al., 2022), we translated the non-English articles into

| Model | Mean $r$ | Max $r$ | Rank |
|---|---|---|---|
| *Our Model* | 0.860 | 0.861 | - |
| HFL | 0.839 | 0.872 | 1 |
| EMBEDDIA | 0.704 | 0.855 | 2 |
| L3i | 0.786 | 0.855 | 3 |
| WueDevils | 0.822 | 0.857 | 4 |
| DataScience-Polimi | 0.770 | 0.873 | 5 |

Table 1: Our model compared to the top five models in the English-only portion of the SemEval news comparison task, including those with the top mean (HFL; Xu et al., 2022) and max (DataScience-Polimi; Di Giovanni et al., 2022) Pearson correlation. Our bi-encoding strategy is adapted from the design of the WueDevils model (Wangsadirdja et al., 2022), though our refinements lead to better performance. Using the scoring metric from the SemEval competition, our system would rank 2nd of 30, and achieves the overall best mean performance (computed over random seeds).

English using the Google Translate API, keeping the human similarity ratings for the corresponding non-English article pairs.

Our model was trained over 2 epochs with a batch size of 5 and a linear learning rate starting at 2e-6. On the English subset of the SemEval test set, our model achieves near state-of-the-art performance, as shown in Table 1.[3] Appendix A includes additional experiments testing the impact of various modeling choices, and our best-performing model is available online.

## 3.3 Creating Story Clusters

To create story clusters, we first embedded each article in our complete news dataset using our fine-tuned article encoder. We then computed the similarity of article pairs to identify story clusters. To do so, we first reduced the search space of potential article clusters by considering only article pairs published less than eight days apart, with at least one named entity in common. Named entities that were very common (found in over 20,000 articles) were discarded from this filtering process in order to maintain computational tractability (details in Appendix B).

For the remaining article pairs, we computed their cosine similarity and kept only the article pairs having a cosine similarity greater than 0.9, based on pilot experiments. The resulting pairwise similarity matrix of all article pairs over our threshold forms a graph, and our story clusters are the connected components of this graph.

---

[1] The original SemEval training set included 4,918 articles, but a slightly larger number of article pairs was actually released online and used by other teams in the task. The annotations include similarity judgements on seven different dimensions, but we only used the OVERALL similarity judgements, which we linearly rescale to the range [0, 1].

[2] https://huggingface.co/sentence-transformers/all-mpnet-base-v2

[3] Averaging over random seeds, our model achieves the highest mean correlation score, but would rank second due to how the competition scoring weights multiple entries.

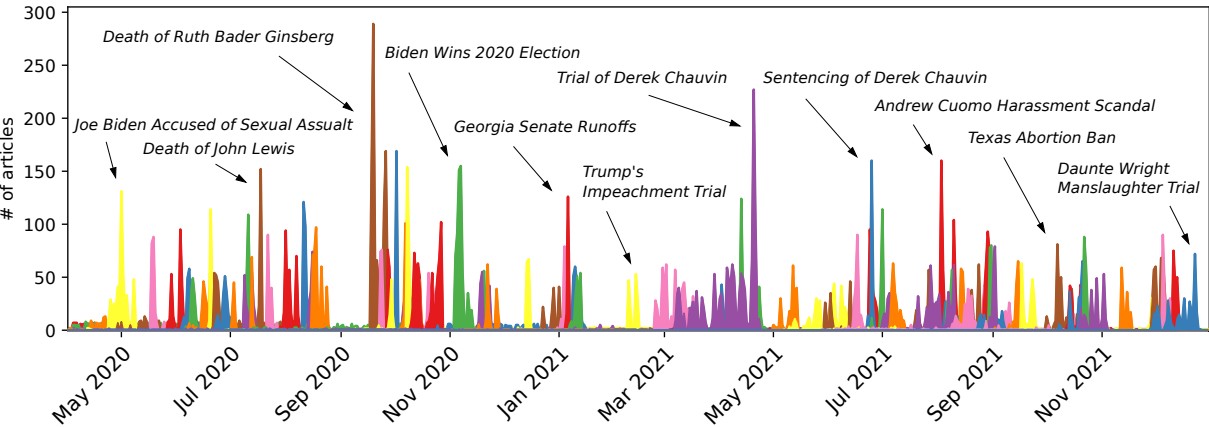

Figure 2: Time series of all 98 media storms from April 1, 2020 – December 31, 2021, as clustered based on our article similarity model (§3.2). Stories are classified as media storms using criteria based on cutoffs for duration and number of outlets with a high volume of sustained coverage (see §3.3).

## 3.4 Identifying Media Storms

Our corpus of story cluster labelled news articles lends itself naturally to the identification of media storms at the level of events. We adapted Boydstun et al. (2014)'s definition of media storms toward this end.[4] Working with the notions of a high volume of sustained coverage across multiple outlets for a sustained period of time, we operationalize "storm mode" as periods in which an outlet devotes at least 3% of their coverage to a story in a three day period.[5] Media storms are those stories for which coverage lasts at least one week and where at least five outlets cover the story in this mode. All articles clustered as part of that story cluster are considered as part of the overall media storm. These criteria yield 98 media storms in our 21 month period. Figure 2 shows the timeline of all storms with example storm descriptions. Brief descriptions of all storms can be found in Table 4 in Appendix D.

## 4 Characterizing Media Storms

Our definition and selection process focuses on media outlets reporting in storm mode over a substantial period of time. We now characterize which

| Feature | Min | Max | Mean | Median |
|---|---|---|---|---|
| Number of Articles | 51 | 1378 | 207.5 | 156 |
| Duration (days) | 7 | 54 | 15.0 | 11 |
| Number of Outlets | 5 | 200 | 76.1 | 74 |
| % National | 0 | 100 | 59.1 | 62 |

Table 2: Descriptive statistics of our 98 media storms where % National refers to the percentage of articles from national outlets. Summary statistics indicate that media storms identified using our pipeline are inline with expectations from prior literature.

story clusters are selected based on this definition. Our summary statistics of media storms serve two ends: first, they confirm that our selection criteria yield storms that are in line with previous theoretical and empirical work on media storms. Second, they demonstrate that previous hypotheses about the quantitative features of news storms have empirical support in our data.

### 4.1 Storm Duration

As shown in Table 2, our media storms have an average article count of 207.5 and an average duration of 15 days. Prior work has posited that media hypes die down after 21 days and, in analyses of articles from two outlets, presented empirical evidence of storms having an average duration of between 14.5 and 16.7 days (Vasterman, 2005; Wien and Elmelund-Præstekær, 2009; Boydstun et al., 2014). Our average duration of 15 days aligns with these expectations from prior literature and confirms that our cutoffs in storm selection are reasonable.

### 4.2 Storms as Multi-Outlet Phenomena

In addition to characterizing storm lengths, prior work has also hypothesized that media storms in-

---

[4]Boydstun et al. (2014) define media storms as "an explosive increase in news coverage of a specific news item constituting a substantial share of the total news agenda [across multiple outlets] during a certain time." Because they were relying on hand-coded headline topics to identify storms, and only considered two individual outlets, they ignored the "multi-media-ness" part of their definition, and operationalized this concept as at least a 150% increase in coverage of an issue, lasting at least a week, and comprising at least 20% of a paper's coverage devoted to the issue during that time. Because we identify storms at the level of *events*, rather than issue areas, we use somewhat different criteria.

[5]We exclude three day windows in which an outlet has fewer than 40 stories.

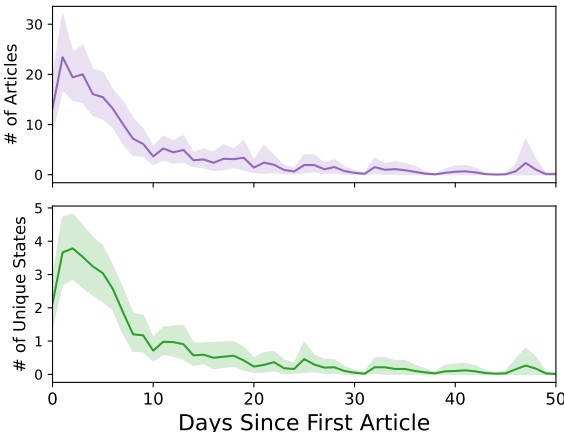

Figure 3: Average time series for the daily article count (top) and unique US state count (bottom) of media storm coverage. The average storm peaks early and then fades slowly from the media. Shaded bands correspond to bootstrapped 95% confidence intervals.

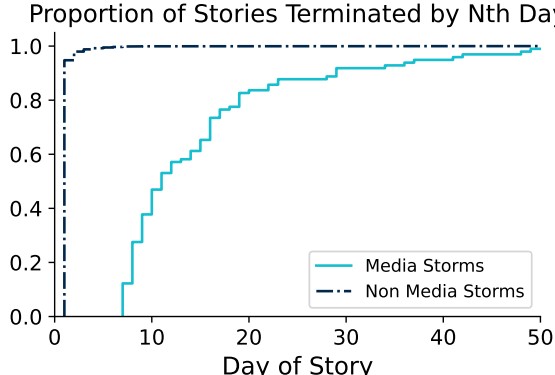

Figure 4: Empirical CDF of the duration of media storm and non-storm stories. Media storm stories last far longer than the average non-storm story.

volve coverage in many outlets (Boydstun et al., 2014). Given that our storm selection criteria require only five outlets to report in storm mode, it is entirely plausible that only a few outlets in a niche corner of the media ecosystem could generate storms. Summary data of our media storms (Table 2), however, confirms that this is very rarely the case, with the average storm involving 76 outlets. Furthermore, storms are far less likely to be contained within one part of the media ecosystem than non-storms, with the average storm being comprised of 59.1 and 40.9 percent national and local coverage respectively.

### 4.3 The Timeline of Storm Development

The question of how quickly information in media ecosystems rises and falls is the subject of much prior work (e.g., Wien and Elmelund-Præstekær, 2009; Aldoory and Grunig, 2012; Castaldo et al., 2022). Time series analyses of political rumors, misinformation, memes, and search keywords have found that attention around a given construct displays distinct patterns, typically rising to a peak and falling very quickly thereafter (Leskovec et al., 2009; Shin et al., 2018). Media storms, however, have been differentiated as having an explosive rise in coverage followed by relatively smooth, gradual changes (Boydstun et al., 2014).

To elucidate the general temporal dynamics of media storms, Figure 3 (top) displays the average volume of coverage over time across all 98 storms.

This average series quickly rises to prominence within three days, declines steadily until day 10, and diminishes far more gradually after that. A similar trend is observed for the number of unique US states represented by the outlets involved in a storm's coverage (Figure 3, bottom).

Temporal statistics of media storms provide support for previous findings that they are both explosive and long-lasting. The median day for a storm's peak coverage is day 4.5, and the most common day for peak coverage is the first day, followed by days 2–5.[6] In other words, most storms peak almost immediately, demonstrating the explosiveness of their coverage. To show how long media storms linger in the press, we use the empirical CDF of storm duration, which is presented in Figure 4. Here, we find that the percentage of storms lasting for weeks of coverage is far higher for storms than non-storms.

Although most storms have an early peak of coverage, we also identify heterogeneity in media-storm development, with a significant fraction of stories having two distinct peaks or even peaking at the end of their respective time series. These late-peaking stories appear to relate to events that are inherently ongoing or can be anticipated.

To verify this qualitatively, we perform a manual review of storms peaking on or after day 15. Out of these 13 late-peak storms, eight cover stories with a clear anticipated peak event such as court cases, elections and bills passing through Congress. The remaining five storms pertain to ongoing coverage of COVID-19 and assault allegations made by Tara Reade during Joe Biden's campaign. This aligns

---

[6]The distribution of peak timing and other properties of our media storms are included as plots in Appendix C.

with work correlating the temporal dynamics of the news with stories pertaining to different topics (Geiß, 2018), and indicates a need for further research to understand how storms with different content develop. Additional plots showing the distribution of storm properties are included in Figure 10 in Appendix C.

## 5   Topical Distribution of Storm Coverage

Examining media storms at the level of issues, Boydstun et al. (2014) hypothesized and found that media storms tend to be more skewed towards certain issues than overall media coverage, with the majority of storms being related to Government Operations, Defense, or International Affairs (which were also the most common issues in overall coverage, but to a much lesser degree).[7] This observation also aligns with literature positing that stories with certain news values are more appealing to the media and therefore more likely to trigger storm-development mechanisms (Harcup and O'Neill, 2017).

To test this theory at the level of media storms tied to specific events, we ran Latent Dirichlet Analysis (Blei et al., 2003) on our entire corpus of news articles using Mallet (McCallum, 2002). We set the number of topics to be 30, infer a distribution over topics for each article, and associate each article with its highest-scoring topic. Finally, we aggregate these topics across storm and non-storm articles.

As shown in Figure 5 and supplemental Figures 11 and 12 in Appendix C, we confirm the previous finding that media storms have a topic distribution which is far more skewed than the topic distribution for non-storm articles. In addition, we find that certain topics make up a much larger fraction of coverage for storms than non-storms. In particular, topics related to politics, natural disasters, and policing are overrepresented in storms, whereas topics related to sports, entertainment, and daily life are underrepresented. In contrast to Boydstun et al. (2014), however, we find that topics related to defense and international affairs are less common among storms compared to non-storms, as well as being less common overall, compared to other topics (see Figure 12 in Appendix C). This may relate in part to our use of a much wider set of media outlets, including many local news sources.

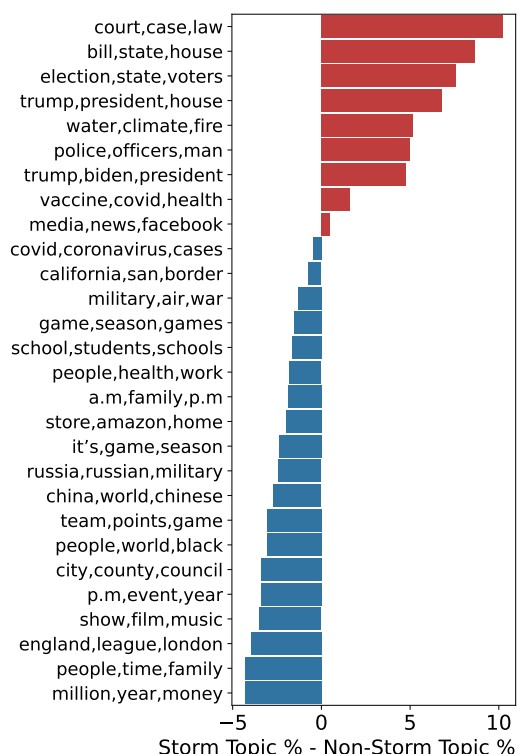

Figure 5: The difference in the percentage of articles associated with a given topic between articles in media storms and those not in media storms. Articles are assigned the topic with the highest proportion in their topic distribution. Media storms involve disproportionately more coverage of politics, court cases, policing, and climate.

Regarding the topic skew of media storms, we hypothesize that this distinction may be a result of certain types of events lending themselves to ongoing and follow-up coverage. A topic like entertainment may contain more disparate one-off events, whereas legislation and disasters reflect events whose development and consequences can last for longer periods of time.

## 6   Mechanisms of Storm Development

Having provided evidence for media storms as a distinct temporal and topical phenomenon, we turn toward testing two mechanisms associated with storm development. First, we test whether storms are associated with a lowering of the gatekeeping threshold—an increased recognition of the newsworthiness of a particular topic. Second, we test whether media storms are associated with intermedia agenda-setting behavior, a phenomenon wherein outlets mimic one another to keep their coverage relevant.

---

[7]Boydstun et al. (2014) used 27 hand-coded issue areas from the Comparative Agendas Project.

## 6.1 Gatekeeping

Gatekeeping refers to the process of filtering or selecting what information to present out of the vast universe of possibilities (Shoemaker and Vos, 2009). One potential explanatory mechanisms for the emergence of media storms is a lowering of the gatekeeping threshold. According to this theory, once a particular topic, issue, or event has received a critical mass of coverage, it is seen as more newsworthy and will therefore receive even more coverage (Boydstun et al., 2014). At the news-outlet level, this effect can be explained by disruptions in publishing routines. Indeed, interview data finds that in extreme cases, news outlets may even reassign almost every journalist to a particular issue or event (Hardy, 2018).

Of interest here is whether a surge in attention on a single story is correlated with a broader shift in the attention devoted to related stories. According to this framework, once the gatekeeping threshold is lowered, events are stitched together by the media to form a broader reporting agenda with even historical events being introduced back into the discourse (Kepplinger and Habermeier, 1995; Seguin, 2016; Baumgartner and Jones, 1993). It follows that if the gatekeeping threshold mechanism holds for our media storms, they should be accompanied by waves of additional topically related coverage.

To test this hypothesis empirically, we first identify a time period of 14 days before and after the first day of coverage of each storm. We then limit ourselves to only outlets that report on the media storm in question. For each day in our 29-day window, we then measure the mean percentage of coverage devoted to the topic associated with corresponding storm, averaged across all storms.[8]

When including all articles, we find an increase of 1.6 percentage points from the mean pre-storm level to the maximum level during the storm (see Figure 6). If we exclude the articles associated with the media storm, and only consider others on the same topic, we find there is still an increase of 1.1 percentage points from the mean pre-storm topic level to the maximum post-storm level. Together, these suggest that media storms lead to increased attention to the broader issue they are part of, thereby disrupting normal publishing routines.

Perhaps more surprisingly, we also see that at-

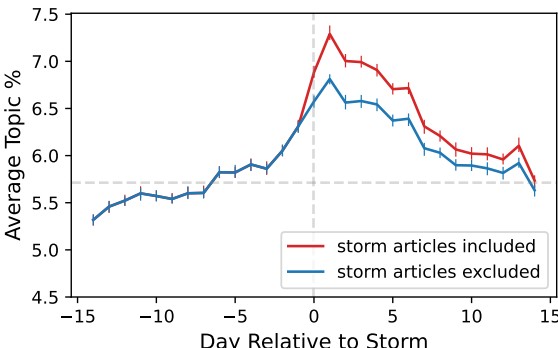

Figure 6: Time series depicting the average percentage of each media storm's associated topics in articles before and after the storm's first day. Each media storm is assigned the topic with the highest percentage when summing over all of its articles. Media storms are preceded by a rise in their associated topic and this topic increases significantly once the storm begins.

tention to issues begins to rise before the start of associated storms. Some of this might be accounted for by early articles that were not properly matched to the story cluster, but a more compelling explanation is that coverage of an issue begins to rise early in anticipation of a particular event. Given that topics related to somewhat predictable events, such as court cases, are over-represented among storms (see Figure 5), this early rise may represent news organizations "priming the pump" in advance of their coverage of the actual event.

## 6.2 Intermedia Agenda Setting

A second mechanism proposed to explain media storms is inter-media agenda setting (Boydstun et al., 2014; Hardy, 2018). According to this theory, reporting from one news outlet signals the importance of a given event or issue to other outlets (Vliegenthart and Walgrave, 2008).

To determine which outlets lead in storm coverage, we analyze the patterns of publishing on storms across outlets. To do so, we construct a weighted graph with outlets as nodes. For all outlets and storms, we find the first article each outlet publishes about a given storm. We then assign credit for influence to all outlets that had an article related to that storm in the preceding two days. The edge between outlet $i$ and outlet $j$ is thus the number of storms in which outlet $i$ published in the two days preceding the first article from outlet $j$.

Using the 20 outlets with the most articles in our 98 storms (among those rated as reliable in the NELA data), Figure 7 shows the directed influ-

---

[8]A storm's topic is defined as the topic with the highest concentration when summing topic distributions across all of the associated articles.

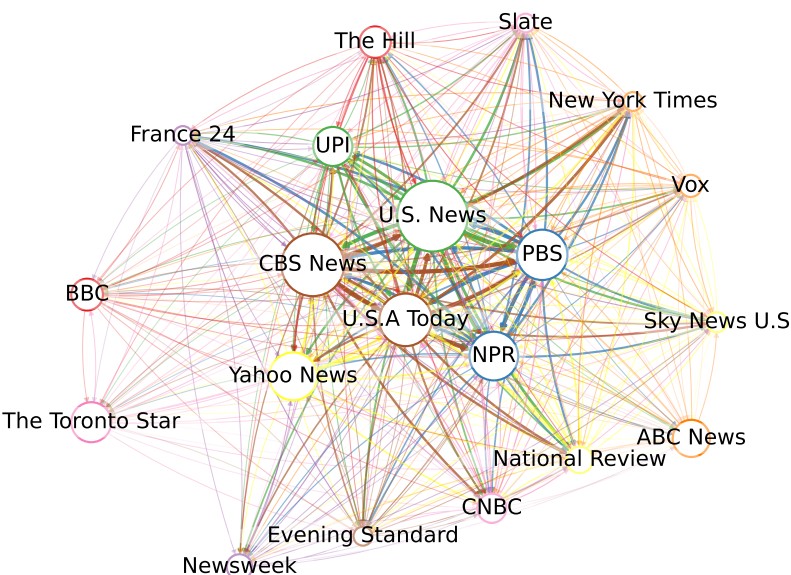

Figure 7: A network depiction of influence among the top 20 national news sources rated as reliable in our data. Directed edges show amount of influence based on timing of reporting on storms. Larger nodes have higher net influence (total outgoing minus incoming edge weights). Colors are used only to differentiate sources.

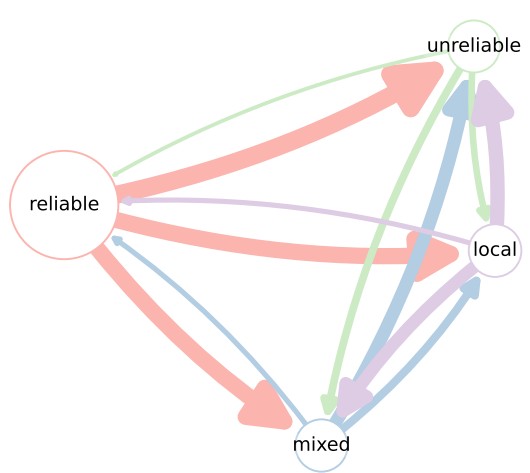

Figure 8: A network depiction of influence between all local outlets and national outlets having three different factuality ratings (reliable, unreliable, mixed). Only the top 200 outlets in terms of media storm coverage are considered. Edge weights correspond to the amount of influence between different types of outlets, and node sizes represent net influence, as in Figure 7.

ence among these outlets, with node size mapping to the net influence of that outlet, considering all incoming and outgoing edges. This graph is consistent with the hypothesis that intermedia agenda setting is present as storms migrate through the media ecosystem. Overall, we find a rich network of connections, with no single outlet consistently publishing in advance of all others. Nevertheless, a few

outlets do appear most prominently in this analysis, including nonprofits (PBS, NPR), and digital-only platforms like Yahoo News.

Figure 8 presents a similar analysis at the level of different parts of the media ecosystem. Once again, we see a clear distinction that aligns with an intermedia agenda setting mechanism for storm development. In particular, reliable news appears to be highly influential on local and on less-reliable national news. The opposite is true of unreliable news, which has weak influence relationships to all outlet types. One finding of particular interest is that local news is disproportionately influenced by reliable news but influences mixed and unreliable news disproportionately. The discovery of this heterogeneity in influence signals a need for more research on the relationship between local coverage and national coverage of varying types.

## 7   Discussion

When considered in the context of prior work, our results have implications for the media's effect on public and congressional agendas. In their original paper, Boydstun et al. (2014) find that Google queries corresponding to a given event spike more during media storms than during coverage of similar non-storm stories. Furthermore Walgrave et al. (2017) find that news stories exert a stronger influence over the congressional agenda when these

stories are part of a media storm. By characterizing the temporal dynamics, topical skew, and lead-lag relationships associated with media storms at the level of events, we give context for what types of stories have a larger agenda setting effect and where these stories originate from.

Although our analysis of media storms revealed a somewhat different distribution of topics than the hand-coded analysis of news headlines in Boydstun et al. (2014), overall our characterization of storms broadly matches theirs. Some events, such as accusations of sexual assault against Joe Biden, trigger media storms involving hundreds of news articles across dozens of outlets. Others, such as airlines cancelling flights due to staffing shortages, have a much smaller footprint, but still persist in the news cycle for more than a week.

While some storms, (such as the trial of Derek Chauvin), peak weeks after they begin, the typical storm follows a pattern of explosive coverage (peaking within the first few days), followed by a gradual decline in attention. Moreover, this initial increase in coverage is accompanied by, and sometimes even anticipated by, an increase in attention to the broader issue area that the storm is part of (e.g., policing, natural disasters, etc.). All of these findings point to the operation of underlying mechanisms governing news production, with limited resources available within each news outlet.

Beyond the results in this paper, the methodology, model, and dataset we contribute open the door for future research on media storms. One important direction is the causal investigation of media influence. For example, identification of matched pairs of media storms with comparable non-storms would allow for more direct testing of the key conditions which cause stories to gain traction. Such a study is difficult because many media storms may have no comparable counterfactual non-storm, and the relevant features needed to match pairs may be unobserved. Unanticipated real world events (e.g. natural disasters) may also provide natural experiments to study the media conditions in which storm are most likely to develop, and could be compared to storms based around events which can be anticipated. Alternatively, follow up studies could investigate how interactions between traditional media, social media, and political elites influence the trajectory of media storms.

In addition to the estimation of larger and more diverse diffusion networks, future work could also connect these results with the literature on framing and agenda setting. Our analysis finds that unreliable media frequently lags behind other media sources; however, it is unclear why this is the case. Past work has hypothesized that alternative media frequently frames itself in opposition to the mainstream, revealing the "truth" that the mainstream is "unwilling" to say (Andersen et al., 2023; Holt et al., 2019). If the framing of outlets' coverage can be connected to patterns of diffusion, this would result in a far more complete picture of why media outlets report what they report when they do.

Finally, future work may address the limitations of our news similarity model and article matching pipeline. In terms of efficiency, a faster article matching approach would allow for analysis of the much larger, global media ecosystem. In terms of accuracy, our model's performance is typically lower on articles generated from templates such as COVID-19 trackers. A more sophisticated iteration would reliably distinguish between articles about different events in spite of their high textual similarity.

# 8 Conclusion

News organizations shape the public perception of what issues are most salient and media storms play an important role in saturating coverage on a single story. Here, we introduce a new computational approach to quantifying storms and their influence across nearly two years and 4 million news articles, using a new state-of-the-art model to identify news articles on the same event.

Using the 98 storms in our data, we show how they migrate through the media ecosystem via structured lead-lag relationships between outlets and how their development is associated with increased interest in the broader topic or issue that the storm is about. Taken together, our results distinguish media storms as a phenomenon with distinct temporal and topical characteristics. Furthermore, they indicate that some news outlets and categories of news are seemingly more influential than others. These findings give insight into how large news stories form and spread, providing a stepping stone for further computational investigation.

## Acknowledgements

We would like to think Amber Boydstun, Jill Laufer, and anonymous reviewers for their helpful feedback and suggestions.

## Limitations

Our study uses state-of-the-art modeling to present a large scale empirical investigation of media storms across many outlets with diverse audiences. However, there are a number of notable limitations to our approach as well as directions for future research.

One initial limitation in our approach comes from our reliance on three existing datasets of news articles. These datasets only cover a 21 month overlapping period and include a variety of outlets, some of which are based outside the United States. Given this composition of outlets, our data is likely not representative of any one news ecosystem but rather captures a broad sample of the U.S. national and local media outlets with some international coverage included.

In addition, our three news datasets were originally collected from RSS feeds, meaning that we have limited knowledge on the completeness of coverage. While RSS feeds typically mirror the online feed of news organizations, outlets which push only a fraction of coverage to their feed may interfere with the accuracy of our results. Despite these potential issues, we find that the duration and temporal patterns of our identified media storms align with what is expected given prior literature.

In terms of identifying storms, one limitation in our work is the scalability of our clustering pipeline. To avoid computing pairwise similarity among $n^2$ pairs for millions of articles, our current approach considers pairwise similarity between all articles with named entity pairs having under 20,000 occurrences, restricted to those article pairs that are published less than eight days apart. While larger events can be identified through links between pairs of articles with different, less common named entities, our cutoff of 20,000 might unintentionally exclude some of the largest news storms. Most of the entities that were removed correspond to prominent politicians or organizations (e.g., Joe Biden, the FBI, etc.). However, one notable exception is that our dataset does not include a media storm specifically about the death of George Floyd, as this name was mentioned in more than 80,000 articles in our full dataset, and was therefore removed before forming story clusters.

When it comes to analyzing our identified media storms, our evidence has important limitations as well. Our results on the development of media storms are merely correlational and fail to assess what causes media storms. Given the self-referential nature of the news production process (Kepplinger and Habermeier, 1995), these causal mechanisms are incredibly difficult to locate. As such, large scale descriptive results still represent an important step in extending the current state of the literature.

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

## A News Similarity Modeling Details

The SemEval task on news article similarity (Chen et al., 2022) was a multilingual challenge, where teams were given access to pairs of news articles rated in terms of multiple types of similarity (e.g., temporal, geographic, tone) on a scale of 1 to 4. We relied on the "overall" similarity judgements, and rescaled the similarity scores to $[0, 1]$.

The top-performing model in the competition (Xu et al., 2022) used a cross-encoder, in which a model was trained using pairs of article texts given to the model simultaneously. Because such an approach does not scale to rating the similarity of millions of articles, we opted to use a more efficient bi-encoder (as used by other submissions, such as Wangsadirdja et al., 2022), in which a model is trained to usefully embed articles, such that the similarity of embeddings can then be quickly computed.

In developing our model, we performed an ablation study to evaluate the effectiveness of various data augmentation decisions. Taking inspiration from (Xu et al., 2022)'s approach, we experimented with translation of training data into English as well as various concatenation of the head and tail of news articles. Five-fold cross validation was used to evaluate the efficacy of these approaches, without making any use of the test data. When using translated data, only the English article pairs from a given fold were used for evaluation.

As shown in Table 3, both methods appear to have a minimal effect on the performance of our model. Despite this null result, we still use translated data and a combination of 288 head tokens and 96 tail tokens. This choice is made because both augmentation strategies were effective in Xu et al. (2022), which was the top-ranked model when evaluated on the SemEval English-only test set.

## B Details of Article Clustering

In order to reduce the number of pairwise article similarity comparisons needed to augment our dataset of over 4 million articles, we first applied a named entity filter. Named entities were extracted from each article using the named entity recognition pipeline in spaCy's `en_core_web_md` model.[9]

After identifying these entities, we kept only those entities corresponding to an Organization, Event, Person, Work of Art, or Product, as entities

---

[9]https://spacy.io/models/en

---

| Head | Tail | Translated | Mean | Max | S.D. |
|------|------|-----------|------|-----|------|
| 288 | 96 | yes | 0.885 | 0.898 | 0.017 |
| 192 | 192 | yes | 0.886 | 0.894 | 0.006 |
| 384 | 0 | yes | 0.885 | 0.900 | 0.008 |
| 288 | 96 | no | 0.884 | 0.904 | 0.015 |

Table 3: Ablation results showing Pearson correlation coefficients over five-fold cross validation. Translated refers to the inclusion of the back translation of non-English articles. Head and Tail refers to the number of tokens concatenated from the head and tail of articles, respectively. Translated article pairs are always excluded from evaluation.

of other types were more likely to be associated with an unreasonably large number of articles. For computational reasons, entities with over 20,000 associated news articles were also removed. This resulted in the exclusion of 113 entities, which left 2,614,176 distinct entities remaining with at least two associated articles. Finally, we created an inverted index between each entity and its corresponding articles. This index was used to determine which pairs of articles had at least one entity in common and were thus designated as eligible for comparison using our pairwise similarity model.

While entities such as "FDA", "Biden", and "Trump" were among the removed entities, we found multiple media storms with coverage relating to these entities and mentioning them in the article text. This suggests that many media storms are identifiable with entities which are less common across all articles and more specific to the associated event. For completeness, we include the full list of excluded entities in the replication code associated with this project.

When embedding articles using our similarity model, we merged the first 288 tokens of each article including the title with the last 96 tokens, as discussed in Appendix A. As shown in Figure 9 this resulted in the majority of articles being truncated. This effect was only modest, however, with the median article length being reduced from 467 words to 317 words when identifying words by splitting on whitespace in the article text. We also experimented with models having longer context windows but found that this produced inferior results in pilot experiments.

## C Additional Details on Characterizing Media Storms

Figure 10 shows some additional plots describing the distribution of media storms, in terms of total

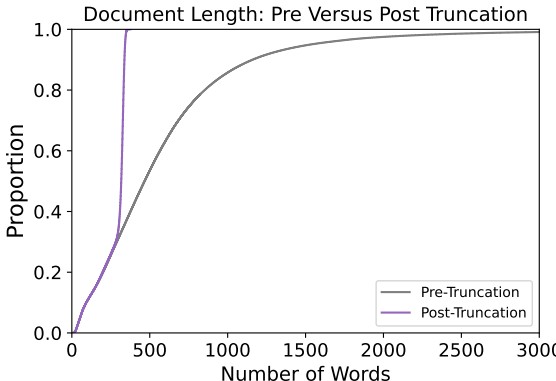

Figure 9: The empirical CDF plots for the lengths of articles before and after truncation. Number of words is determined by splitting articles on whitespace.

article count, peak day of storm coverage, storm duration in days, and percentage of coverage in national outlets (as opposed to local). Most storms are primarily national, but with a small number that are almost entirely local. Similarly, there is a long tail of storms in terms of duration, peak day, and total volume, with the largest storms comprising over 1,000 articles.

Figures 11 and 12 show the individual average topic distributions for both storms and non-storms as a complement to Figure 5 in the main paper. As can be seen, the distribution of storm topics is more skewed, and shows considerable difference in relative topic rankings.

## D   Media Storm Descriptions

A complete list of the media storms found in this work is given in Table 4, which includes start date, date of peak coverage (in terms of article count), duration, total number of articles, percent of coverage that was from national sources, and a brief summary description of the storm.

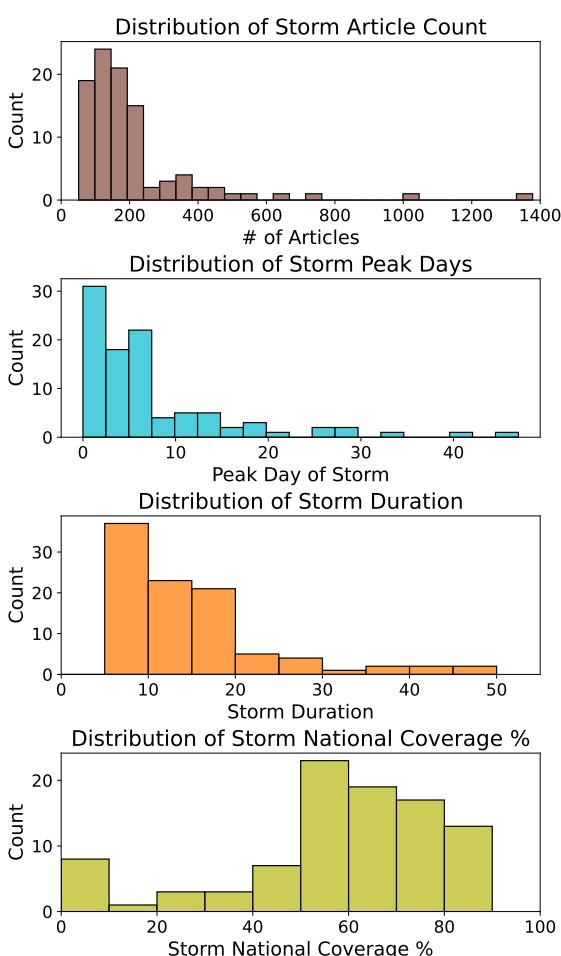

Figure 10: Summary distributions for our media storms.

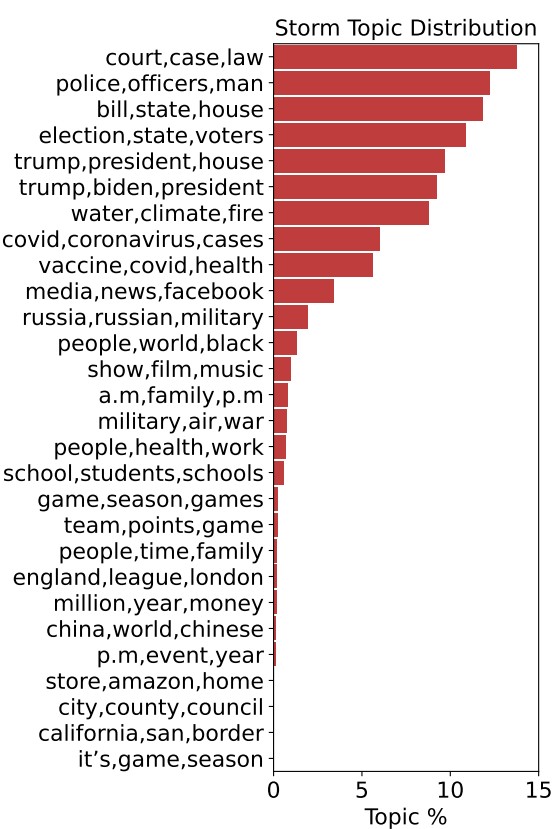

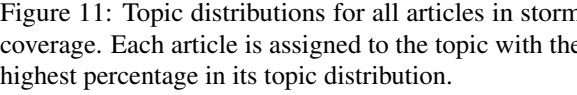

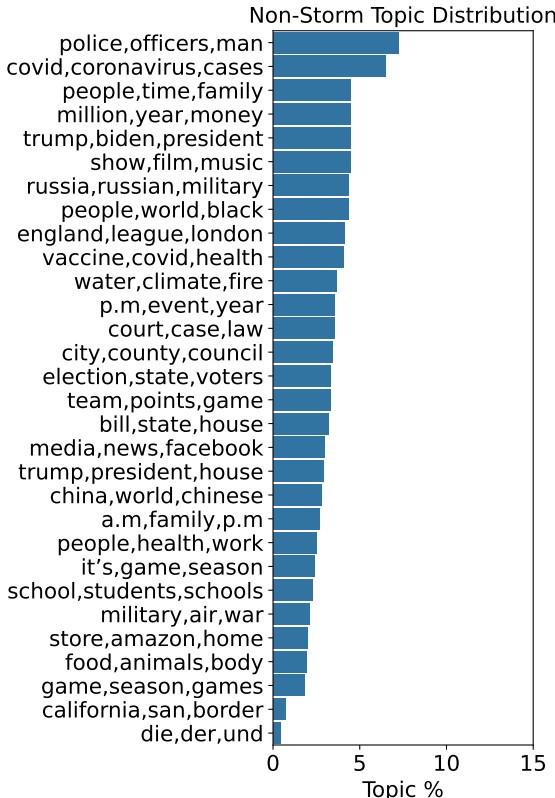

Figure 11: Topic distributions for all articles in storm coverage. Each article is assigned to the topic with the highest percentage in its topic distribution.

Figure 12: Topic distributions for all articles in non-storm coverage. Each article is assigned to the topic with the highest percentage in its topic distribution.

| Start date | Peak date | Length (days) | Article count | % Nat'l | Description |
|---|---|---|---|---|---|
| Apr 05, 20 | Apr 06, 20 | 8 | 91 | 28.6 | Boris Johnson's contraction and recovery from COVID-19 |
| Apr 06, 20 | Apr 23, 20 | 19 | 52 | 0.0 | Minnesota Gov. Walz gives COVID-19 update |
| Apr 12, 20 | May 01, 20 | 34 | 516 | 81.8 | Joe Biden accused of sexual assault |
| Apr 25, 20 | May 06, 20 | 18 | 109 | 52.3 | Dallas salon owner jailed for violating COVID restrictions |
| Apr 28, 20 | Apr 28, 20 | 8 | 114 | 77.2 | Mike Pence criticized for not wearing mask to Mayo Clinic |
| May 05, 20 | May 07, 20 | 9 | 57 | 100.0 | Adam Schiff and "Russiagate" transcripts |
| May 18, 20 | May 19, 20 | 11 | 206 | 65.5 | Response to Trump taking hydroxychloroquine |
| May 27, 20 | Jun 03, 20 | 16 | 251 | 53.0 | Court trials of officers in George Floyd's murder |
| Jun 01, 20 | Jul 12, 20 | 48 | 90 | 1.1 | Decline and subsequent rise of COVID cases in Minnesota |
| Jun 01, 20 | Jun 05, 20 | 10 | 89 | 22.5 | Minneapolis bans police chokeholds in response to George Floyd's murder |
| Jun 04, 20 | Jun 08, 20 | 14 | 162 | 78.4 | Support for defunding, abolishing police in Minneapolis |
| Jun 06, 20 | Jun 10, 20 | 12 | 144 | 70.8 | Renaming of army bases named after Confederate leaders |
| Jun 07, 20 | Jun 20, 20 | 17 | 236 | 78.8 | Trump attempts to prevent Bolton from publishing his memoir |
| Jun 12, 20 | Jun 12, 20 | 10 | 103 | 0.0 | COVID-19 updates in Minnesota, North Dakota |
| Jun 18, 20 | Jun 28, 20 | 15 | 189 | 71.4 | Mississippi to remove Confederate emblem from flag |
| Jun 19, 20 | Jun 20, 20 | 7 | 148 | 67.6 | Top Manhattan prosecutor fired by Trump |
| Jun 21, 20 | Jun 22, 20 | 9 | 177 | 74.6 | Bubba Wallace finds noose in garage; subsequent coverage |
| Jul 07, 20 | Jul 08, 20 | 7 | 93 | 55.9 | Trump pushes states to reopen their schools |
| Jul 08, 20 | Jul 13, 20 | 8 | 193 | 73.1 | Death of "Glee" actress Naya Rivera |
| Jul 10, 20 | Jul 11, 20 | 7 | 222 | 82.4 | Trump commutes Roger Stone's sentence; ensuing backlash |
| Jul 13, 20 | Jul 15, 20 | 17 | 132 | 41.7 | Major retailers update mask policies |
| Jul 29, 20 | Aug 01, 20 | 20 | 316 | 77.8 | Trumps says he will ban TikTok |
| Aug 09, 20 | Aug 11, 20 | 8 | 233 | 59.7 | Joe Biden selects Kamala Harris as running mate |
| Aug 15, 20 | Aug 18, 20 | 12 | 363 | 43.5 | Postmaster general's involvment in mail-in voting controversy |
| Sep 09, 20 | Sep 14, 20 | 9 | 81 | 74.1 | TikTok's owner partners with Oracle rather than Microsoft |
| Sep 17, 20 | Oct 08, 20 | 36 | 442 | 64.3 | Controversy over presidential debate after Trump contracts COVID-19 |
| Sep 18, 20 | Sep 19, 20 | 14 | 1026 | 72.1 | Ruth Bader Ginsburg dies; Trump picks Amy Coney Barrett to replace |
| Oct 04, 20 | Oct 07, 20 | 7 | 188 | 42.0 | Vice-presidential debate 2020 |
| Oct 05, 20 | Oct 07, 20 | 7 | 155 | 61.3 | Hurricane Delta |
| Oct 08, 20 | Oct 16, 20 | 15 | 100 | 98.0 | Controversy surrounding potential for Biden to pack the Supreme Court |
| Oct 10, 20 | Oct 27, 20 | 19 | 622 | 62.4 | Amy Coney Barrett's confirmation hearings and confirmation |
| Oct 27, 20 | Oct 30, 20 | 7 | 81 | 29.6 | Controversy surrounding extension of ballot counting period in Minnesota |
| Nov 03, 20 | Nov 07, 20 | 22 | 737 | 45.9 | Biden wins 2020 election |
| Nov 15, 20 | Nov 22, 20 | 16 | 161 | 77.6 | Trump's challenge to 2020 election results in Pennsylvania |
| Nov 17, 20 | Nov 19, 20 | 8 | 161 | 85.1 | Michigan certifies election results, sealing Biden's win |
| Nov 27, 20 | Nov 30, 20 | 7 | 107 | 76.6 | Biden wins Wisconsin recount |
| Dec 08, 20 | Dec 12, 20 | 7 | 131 | 59.5 | SCOTUS rejects Trump and Texas's attempt to overturn election results |
| Dec 09, 20 | Dec 15, 20 | 12 | 149 | 66.4 | Electoral College casts votes |
| Dec 31, 20 | Jan 04, 21 | 10 | 146 | 84.9 | US request to extradite Julian Assange is blocked |
| Jan 07, 21 | Jan 13, 21 | 7 | 184 | 73.9 | Impeachment attempt after January 6 |
| Jan 08, 21 | Jan 10, 21 | 9 | 212 | 88.7 | Parler removed from app store |
| Feb 01, 21 | Feb 05, 21 | 8 | 109 | 72.5 | Marjorie Taylor Green ousted from her committees |
| Feb 02, 21 | Feb 03, 21 | 49 | 57 | 7.0 | Coronavirus updates in California; Deaths decline |
| Feb 03, 21 | Feb 17, 21 | 23 | 64 | 7.8 | Horoscopes referencing celebrities |
| Feb 08, 21 | Feb 13, 21 | 8 | 152 | 44.7 | Trump's second impeachment trial |
| Feb 24, 21 | Mar 02, 21 | 23 | 525 | 88.0 | Andrew Cuomo accused of harassment |
| Mar 04, 21 | Apr 20, 21 | 54 | 1378 | 50.0 | Trial of Derek Chauvin |
| Mar 24, 21 | Mar 29, 21 | 8 | 107 | 48.6 | Cargo ship blocks Suez Canal |
| Mar 29, 21 | Apr 06, 21 | 16 | 102 | 84.3 | Arkansas bans trans healthcare for youth |
| Apr 13, 21 | Apr 13, 21 | 15 | 220 | 60.9 | Concern over blood clots after Johnson and Johnson vaccine |
| May 01, 21 | May 12, 21 | 16 | 198 | 85.9 | Liz Cheney ousted from house leadership role |
| May 07, 21 | Jun 04, 21 | 42 | 391 | 75.4 | Biden's infrastructure bill |
| May 10, 21 | May 12, 21 | 8 | 138 | 67.4 | Hacking interferes with Colonial Pipeline |
| May 30, 21 | Jun 13, 21 | 17 | 148 | 65.5 | Netanyahu ousted by Isreali coalition |
| Jun 11, 21 | Jun 16, 21 | 8 | 100 | 70.0 | Biden and Putin meet in Geneva |
| Jun 14, 21 | Jun 17, 21 | 11 | 193 | 72.0 | Biden, senate make Juneteenth a federal holiday |
| Jun 20, 21 | Jun 25, 21 | 10 | 210 | 56.2 | Derek Chauvin Sentenced |
| Jun 23, 21 | Jun 28, 21 | 7 | 106 | 56.6 | Extreme heatwave hits Pacific Northwest |
| Jun 23, 21 | Jun 24, 21 | 9 | 240 | 69.6 | Deal reached on Biden's infrastructure bill |
| Jun 25, 21 | Jul 01, 21 | 11 | 176 | 81.8 | Trump organization charged with tax crimes |
| Jul 01, 21 | Jul 05, 21 | 8 | 156 | 62.2 | Tropical Storm Elsa |
| Jul 05, 21 | Jul 07, 21 | 9 | 205 | 49.3 | Miami death toll climbs after condo collapses |

| | | | | | |
|---|---|---|---|---|---|
| Jul 15, 21 | Aug 10, 21 | 28 | 464 | 62.1 | Infrastructure bill passes through senate |
| Jul 16, 21 | Jul 16, 21 | 10 | 88 | 94.3 | Facebook and Biden clash over COVID-19 misinformation |
| Jul 29, 21 | Aug 03, 21 | 12 | 165 | 61.2 | Biden seeks extension for eviction moratorium |
| Aug 03, 21 | Aug 05, 21 | 11 | 143 | 68.5 | Massive California wildfire |
| Aug 03, 21 | Aug 03, 21 | 16 | 390 | 81.0 | Cuomo resigns due to sexual harassment allegations |
| Aug 05, 21 | Aug 08, 21 | 19 | 101 | 70.3 | Wildfires in Greece |
| Aug 09, 21 | Aug 27, 21 | 41 | 344 | 87.8 | Texas, Florida schools clash with governments over mask mandates |
| Aug 18, 21 | Aug 24, 21 | 9 | 164 | 65.2 | Controversy over Biden's Afghanistan withdrawal deadline |
| Aug 20, 21 | Sep 17, 21 | 29 | 82 | 0.0 | COVID-19 updates in Minnesota |
| Aug 23, 21 | Aug 31, 21 | 16 | 225 | 60.9 | Wildfires approach Lake Tahoe |
| Aug 26, 21 | Aug 29, 21 | 8 | 289 | 60.2 | Hurricane Ida |
| Aug 27, 21 | Sep 15, 21 | 22 | 142 | 72.5 | Gavin Newsome wins recall ellection |
| Sep 14, 21 | Sep 15, 21 | 11 | 127 | 91.3 | Controversy over General Mark Milley's communication with China |
| Sep 14, 21 | Sep 17, 21 | 10 | 111 | 66.7 | Wildfires threaten to destroy California's sequoias |
| Sep 16, 21 | Sep 17, 21 | 13 | 198 | 62.6 | Official bodies approve COVID booster |
| Sep 20, 21 | Sep 20, 21 | 19 | 79 | 0.0 | Minnesota COVID-19 updates |
| Sep 30, 21 | Nov 03, 21 | 37 | 132 | 90.2 | Tight governer race in Virginia |
| Sep 30, 21 | Oct 07, 21 | 29 | 332 | 76.8 | Controversy, political response to Texas abortion law |
| Oct 08, 21 | Oct 21, 21 | 15 | 209 | 77.5 | House votes to hold Steve Bannon in contempt over Jan. 6 |
| Oct 10, 21 | Oct 17, 21 | 9 | 54 | 13.0 | Chicago Sky wins first WNBA title |
| Oct 12, 21 | Oct 20, 21 | 16 | 217 | 54.4 | Official bodies approve mixing COVID vaccines and boosters |
| Oct 20, 21 | Nov 02, 21 | 16 | 242 | 52.1 | Official bodies approve COVID vaccine for children 5-11 |
| Oct 21, 21 | Oct 22, 21 | 7 | 184 | 67.4 | Alec Baldwin kills Halyna Hutchins on set |
| Oct 30, 21 | Nov 11, 21 | 14 | 107 | 72.9 | Judge refuses Trump's request to block Jan. 6 records |
| Nov 10, 21 | Nov 16, 21 | 10 | 126 | 46.8 | Court case of Kyle Rittenhouse |
| Nov 18, 21 | Nov 23, 21 | 8 | 51 | 66.7 | "Unite the Right" trial developments and verdict |
| Nov 25, 21 | Nov 26, 21 | 8 | 64 | 81.2 | Lauren Boebert makes anti-Muslim comments, apologizes |
| Nov 28, 21 | Dec 23, 21 | 29 | 369 | 58.0 | Duante Wright manslaughter trial |
| Nov 29, 21 | Dec 05, 21 | 10 | 337 | 83.7 | CNN fires Chris Cuomo for helping brother |
| Dec 05, 21 | Dec 05, 21 | 8 | 201 | 73.6 | Death of Bob Dole |
| Dec 05, 21 | Dec 16, 21 | 19 | 62 | 19.4 | Horoscopes featuring celebrity names |
| Dec 07, 21 | Dec 14, 21 | 10 | 133 | 75.9 | House votes to hold Mark Meadows in contempt |
| Dec 10, 21 | Dec 11, 21 | 9 | 183 | 64.5 | Deadly tornadoes in Kentucky, southeastern US |
| Dec 14, 21 | Dec 20, 21 | 11 | 66 | 54.5 | Coverage of Omicron variant |
| Dec 23, 21 | Dec 27, 21 | 7 | 81 | 35.8 | CDC shortens COVID isolation window |
| Dec 23, 21 | Dec 24, 21 | 9 | 125 | 60.8 | Airlines cancel flights due to COVID staffing shortages |

Table 4: Descriptions of all 98 media storms, sorted in chronological order by start date.