# OpenReview forum: "When it Rains, it Pours: Modeling Media Storms and the News Ecosystem"
_EMNLP/2023/Conference — EMNLP 2023 Findings_

### Official Review · Reviewer_6B36 · 2023-08-04

**Soundness:** 4

**Excitement:**

3: Ambivalent: It has merits (e.g., it reports state-of-the-art results, the idea is nice), but there are key weaknesses (e.g., it describes incremental work), and it can significantly benefit from another round of revision. However, I won't object to accepting it if my co-reviewers champion it.

**Paper Topic And Main Contributions:**

- This paper aims to analyze news media storms, periods of times in the news media cycle where outlets increase their coverage of an event and also more users interact with it. They propose a model based on article similarity to identify the media storms, identifying 98 of them over a period of 21 months. They then run a large amounts of analysis, confirming existing theories about media storms (such as the fact that they tend to focus more on certain topics or that outlets mimic each other during a media storm).
- Specifically, the authors first uses large amounts from the NELA corpus (millions of articles) and a news similarity model to identify clusters that they say are the media storms. They train the article-similarity model using a dataset of ~5K pairs of articles labeled by humans for their similarity. The article-similarity model is a bi-encoder Neural Network model, trained to minimize the MSE between the cosine similarity of the articles embeddings created by the model and the human similarity ratings. They then use the article-similarity model and their definition of media storms to identify the storms.
- Once identified, the authors analyze different aspects of the media storms:
    - Duration: ~15 days
    - How many outlets report on each storm (~76)
    - How media storms develop: explosively, peaking immediately and sometimes having multiple peaks.
    - The topic distributions: Focus on certain topics, in particular politics where war or entertainment is less focused on
    - The way the storm develops: Either how different outlets copy each other (i.e. local outlets copy reliable outlets) or how coverage of topics increases during the storm (suggesting that storms disrupt normal publishing routines).

**Reasons To Accept:**

- This paper is an explicit study on media storms. I agree with the authors that the analysis of media storms is interesting and that the community can gain lots of insight from them.
- While the authors didn’t use a large amount of data to train their article similarity model, they ran it on a large number of articles to create a large dataset. This dataset can be useful for future work to study media storms.
- The analysis done by the authors confirms what readers expect should happen in media storms. However, the fact that this happens over the authors large analysis confirms that the dataset the authors are proposing is solid.
- The paper is well written and easy to follow.

**Reasons To Reject:**

- The main thing I would have liked to see in this paper is some results using the media storms data for a downstream task. Some of these could also be tasks the authors propose, like identifying trends in future media storms based on historical media storms.
    - Instead, this paper mostly just focuses on analyzing the storms (and some of the results are not surprising given the definition of media storms, like the fact that topic coverage increases). It would be cool if the authors also use the storms data for other tasks, or at least discuss how it can be used. Here are some examples I can think of:
        - Can we predict when a media storm is about to happen? When it’s going to end? How long is it going to be?
        - Can we predict which users are likely to be involved in a media storm? Sources? How does their perspectives change over time?
            - Can we predict which outlets are likely to lead the coverage? During a storm, can we determine this based on past storm data?
        - Do some media storms cover more factual information and others spread more fake news? When does each tend to happen?
            - This can be useful if we know a media storm is happening, then maybe we should analyze the data around the media storm more carefully.
    - Apart from the above, the main contribution of this paper is a dataset and an analysis of it, which in my opinion doesn't make the contribution of the paper extremely strong, thus my ambivalent Excitement score.

**Reproducibility:**

4: Could mostly reproduce the results, but there may be some variation because of sample variance or minor variations in their interpretation of the protocol or method.

**Reviewer Confidence:**

4: Quite sure. I tried to check the important points carefully. It's unlikely, though conceivable, that I missed something that should affect my ratings.

---

> ### Author Rebuttal · Authors · 2023-08-29
>
> Thank you for all of your helpful comments, suggestions, and critiques. We are encouraged by your statements expressing that “the analysis of media storms is interesting”, that “the dataset the authors are proposing is solid”, and that our work is “well written and easy to follow”. We also note that our model and data may be used in not only the analysis of media storms, but rather in any context where scholars are interested in articles which are written about the same event, concept, or topic.
>
> Regarding your suggestions for additional analyses that we could have included, we love the number and variety of questions that you think would be interesting to study, and we hope that our work will enable subsequent investigations along these lines, both by formalizing and characterizing the underlying phenomenon, and by releasing our code and data. Unfortunately, given the available space, we had to make hard choices about what to include. While Boydstun et al.’s paper (2014) introducing media storms was published nearly a decade ago, questions surrounding the mechanisms of media storms and their existence across many outlets of different types have been left unanswered. Our study is the first to test these mechanisms empirically, and this felt like the most important first step towards advancing this research agenda. We hope that our detailed characterization of media storms, along with our analyses of gatekeeping and intermedia agenda setting, will help to further research in this area, including of the kinds of questions you suggest.
>
> Finally, although we were not able to address the downstream effects of media storms directly, our characterization of media storms is consequential because it is linked to downstream effects shown in prior research. Boydstun et al. (2014) provide four case studies showing that Google searches are responsive to media storms. Broader follow up work shows that news stories have greater influence on congressional attention when they take part in a media storm (Walgrave, 2017). In conclusion, our work provides evidence for unanswered questions regarding media storms, introduces resources for general communication research, and opens the door to continued work on the effect of media storms on public opinion and politics.
>
> Boydstun, A. E., Hardy, A., & Walgrave, S. (2014). Two Faces of Media Attention: Media Storm Versus Non-Storm Coverage. Political Communication, 31(4), 509–531. https://doi.org/10.1080/10584609.2013.875967
>
> Walgrave, S., Boydstun, A. E., Vliegenthart, R., & Hardy, A. (2017). The nonlinear effect of information on political attention: Media storms and US congressional hearings. Political Communication, 34(4), 548–570.

---

### Official Review · Reviewer_36nG · 2023-08-05

**Soundness:** 4

**Excitement:**

3: Ambivalent: It has merits (e.g., it reports state-of-the-art results, the idea is nice), but there are key weaknesses (e.g., it describes incremental work), and it can significantly benefit from another round of revision. However, I won't object to accepting it if my co-reviewers champion it.

**Paper Topic And Main Contributions:**

The paper is a study on media-storm and news coverage blow-ups, analyzing them in depth, while also proposing a new article similarity model to compute pairwise similarity and create clusters.

**Questions For The Authors:**

A. Why was "3% coverage in a 3 day period" chosen as the margin for "media storms", was there any basis for this selection? Further elaboration on this would be helpful in replicating the research across further data sources.

B. An interesting followup/addition here would be to see how topics evolve from one to another and if media coverage can follow similar trends? Have the authors explored this avenue?

**Reasons To Accept:**

A. In-depth qualitative and quantitative analysis of media storms, what they are, their implications and how they come into being.
B. While popularity clusters and temporal evolution have been studied in depth with respect to social media and news coverage, outbursts are still significant aspects to study for understanding significance of events. This study, dataset and modeling approach have the potential to serve as good, improved baselines for research of this kind.

**Reasons To Reject:**

A. By selecting the first 288 tokens and the last 96 tokens, how are the authors ensuring the true semantic meaning of the articles is being covered rather than context and linguistic fluff use to build up a news article and conclude it respectively? A lot of articles consist of a premise and conclusion which can possibly consist of other related events or anecdotal pieces etc, that might not be relevant to the exact topic at hand. Whereas, some articles are to the point. When trying to pair such articles with a VERY high threshold of 0.9, there is a strong possibility of missed correlations and missed article linkup, which can possibly alter results (even if slightly). It'll be good to have some more depth about the data in terms of average length of articles when data is truncated etc., and how these scenarios are being handled in the study.

**Reproducibility:**

2: Would be hard pressed to reproduce the results. The contribution depends on data that are simply not available outside the author's institution or consortium; not enough details are provided.

**Reviewer Confidence:**

3: Pretty sure, but there's a chance I missed something. Although I have a good feel for this area in general, I did not carefully check the paper's details, e.g., the math, experimental design, or novelty.

---

> ### Author Rebuttal · Authors · 2023-08-29
>
> Thank you for your thoughtful comments and suggestions. We are greatly encouraged by your appreciation of our “in-depth quantitative and qualitative analyses” in addition to “good, improved baselines” for research into news outbursts.
>
> Regarding our choice to only make use of the first 288 and last 96 tokens of news articles, it is true that this does result in the loss of some information. However, the widespread use of the “Inverted pyramid” in journalism means that the “Who, What, When, Where, and Why” of an article are answered within the first few sentences in most cases (Pöttker, 2003). Furthermore, we incorporate article headlines, in addition to body text, which helps ensure that the most important information is included. Finally, the tradeoff of which parts of an article to use was explored by Xu et al., 2022, who achieved best results when keeping most of the head of the article, with some of the tail.
>
> We also appreciate your concern that a change in our article similarity threshold could alter our results. On this note, we reiterate our finding that the average storm duration of 15 days aligns with prior literature, which gives us confidence that our choices in storm selection are reasonable (lines 271-273). We acknowledge that 0.9 is a high threshold, but any such threshold represents a tradeoff between precision and recall. We emphasize precision because the inclusion of many article clusters which are not about the same event is more likely to muddle the behavior of media storms than analysis of a set of articles which have high similarity but may be incomplete.
>
> Regarding our criteria which requires 3% of coverage in a 3 day period, we acknowledge that different choices might have led to slightly different findings. However, preliminary analyses of the distribution of coverage suggested there is a natural break between stories which are given less than this amount of coverage and those that are given more. Indeed, prior work suggests that we would expect somewhat of a binary distinction in article coverage, based on underlying newsroom mechanisms (Hardy, 2018). We would also like to reemphasize that we will release all code and data required to recreate our analyses, which will enable others to explore the effect of varying choices with respect to such thresholds, including those above.
>
> Finally, we love your idea to explore “how topics evolve from one to another and if media coverage follows similar trends”. We appreciate the suggestion of this exciting research direction and underscore that the provision of our dataset, model, and methodology allow for scholars with a range of backgrounds and expertise to pursue questions such as these in future work.
>
> Pöttker, H. (2003). News and its communicative quality: The inverted pyramid—when and why did it appear? Journalism Studies, 4(4), 501–511. https://doi.org/10.1080/1461670032000136596
>
> Xu, Z., Yang, Z., Cui, Y., & Chen, Z. (2022). HFL at SemEval-2022 Task 8: A Linguistics-inspired Regression Model with Data Augmentation for Multilingual News Similarity. Proceedings of the 16th International Workshop on Semantic Evaluation (SemEval-2022), 1114–1120. https://doi.org/10.18653/v1/2022.semeval-1.157
>
> Hardy, H. (2018). 5. the mechanisms of media storms. In Peter Vasterman (Ed.), From Media Hype to Twitter Storm (pp. 133–148). Amsterdam University Press.

---

### Official Review · Reviewer_iVoD · 2023-08-07

**Soundness:** 3

**Excitement:**

3: Ambivalent: It has merits (e.g., it reports state-of-the-art results, the idea is nice), but there are key weaknesses (e.g., it describes incremental work), and it can significantly benefit from another round of revision. However, I won't object to accepting it if my co-reviewers champion it.

**Paper Topic And Main Contributions:**

Authors combine NELA-GT-2020,  NELA-GT-2021 and NELA-Local datasets and extract 98 media storms from those. They define media storm as shifting from routine. coverage to an intensive focus on a particular event, issue, or topic over a period of time. Using this dataset, they create News Similarity model using bi-encoder MPNet which achieves the best mean Pearson correlation, but second best max Pearson correlation. They create event clusters by using time, NER, and cosine similarity. Then, they define rules to extract media storms and then evaluate different properties of those storms and show they agree with what is already know in the literature.

**Questions For The Authors:**

1. Table 2 shows huge differences among the storms properties. It would be very beneficial for your article to explore more subclusters among these storm types and give us better understanding of the different types of storms and their properties. Figure 5 is a step towards it, but it was hard for me to understand what is x-axis in Figure 5 and what do red and blue colors represent.

**Reasons To Accept:**

1. There is a lot of work done on understanding media storms which agrees with existing knowledge of media storms
2. There is a detailed recipe on how to extract the storms from the news articles

**Reasons To Reject:**

1. There is no novel method
2. There is no novel conclusion or observation
3. Many conclusions are just one of the possible explanations - I suggest authors to focus on finding experiments to show that conclusion they think is more likely is the right conclusion

**Reproducibility:**

3: Could reproduce the results with some difficulty. The settings of parameters are underspecified or subjectively determined; the training/evaluation data are not widely available.

**Reviewer Confidence:**

4: Quite sure. I tried to check the important points carefully. It's unlikely, though conceivable, that I missed something that should affect my ratings.

**Typos Grammar Style And Presentation Improvements:**

- Table 1 has ranks 1, 2, 4, 5. Looks like authors excluded some comparison model and forgot to update the numbers.

- Figure 5 should be redone - it's hard to understand what is the meaning of the colors and of the X-axis.

---

> ### Author Rebuttal · Authors · 2023-08-29
>
> Thank you for your helpful comments and suggestions. We appreciate you noting the amount of work we have put into understanding media storms, and our inclusion of a detailed recipe for extracting them, which underscores the soundness and reproducibility of our work.
>
> In your review, you raise concerns about lack of novelty in terms of methods and findings; respectfully, we would like to challenge both of these points. Regarding methods, it is true that we borrow and combine ideas from past work on predicting article similarity. However, existing methods (which we compare against), are excessively computationally demanding, and were not designed to be applied at scale. The model we have created (and will release for others to use), allows for highly scalable pairwise comparisons (which we apply to millions of articles in this paper), while nevertheless achieving performance on par with the best of existing systems.
> With respect to our results and conclusions, many of the findings we present were only theorized in prior literature, without being experimentally tested—particularly the theories on intermedia agenda setting and gatekeeping. Our results provide the first large-scale empirical evidence for these hypotheses, which is a novel contribution to the literature on media storms.
>
> Regarding your concern that “many conclusions are just one of the possible explanations”, we agree about the importance of this. However, establishing causality is challenging, as there are multiple mechanisms at play and no data to support testing each individual. Thus, fully quantifying the causal mechanisms is beyond the scope of our paper. Nevertheless, the results we present here, along with our characterization of media storms, are an important first step in that direction by showing the media storms effect is real. As discussed on lines 502-510, determining causal mechanisms is an important future direction that requires an experimental design (matching, natural experiment, etc…) taking causality into account. We hope to pursue this line of work in a future paper now that we have confirmed the effect.
>
>
> An additional suggestion was to explore “different types of storms and their properties”. While we agree that there is more we could do in this direction, we refer you to lines 331-338, which focus on the subset of storms which reach peak coverage at the end of the storm rather than the beginning. Here, we find that late-peaking storms appear to cover issues such as court cases, elections, and COVID-19.
>
>
> Finally, we greatly appreciate your technical notes on Figure 5 and Table 1. Your confusion about Figure 5 is a result of a plotting error
> on our part, which resulted in a miscalibrated x-axis. In the corrected figure, (in which 0 occurs at what is currently marked -2), it is obvious that red and blue refer to those differences which are greater and less than zero, respectively. We will fix this figure in revisions.
> In the case of Table 1, we report the performance of our model, along with relevant models from the SemEval task. To clarify, we did not accidentally omit a model, as you suppose; as explained in the Table caption, we included HFL for having the highest mean score, DataScience-Polimi for having the highest max score, and WueDevils because we make use of their bi-encoding strategy in our model. We will further clarify this in the updated caption.
>
> Boydstun, A. E., Hardy, A., & Walgrave, S. (2014). Two Faces of Media Attention: Media Storm Versus Non-Storm Coverage. Political Communication, 31(4), 509–531. https://doi.org/10.1080/10584609.2013.875967

---

### Meta-Review · Area_Chair_o9bP · 2023-09-23

**Recommendation:** 4

**Metareview:**

This paper presents a method to detect and analyze news media storms, which are periods of intense and focused coverage of a certain event, issue, or topic. The paper uses a large corpus of news articles from various sources and a neural network model to measure the similarity between articles. The paper then clusters the articles based on their similarity, time, and named entities to identify the media storms. The paper also examines the characteristics and dynamics of the media storms, such as their topics, duration, diversity, and inter-media influence. Overall, the paper makes contribution towards proposing a scalable news similarity detection model and the availability of the resulting dataset can help subsequent future works on media storms.

I thank the reviewers for providing useful suggestions and the authors for providing relevant responses to the concerns raised. I encourage the authors to incorporate these discussions in the future versions of the paper, such as, providing the rationale for using particular portions of the documents for similarity computation or for the specific choice of parameters for determining media storm, apart from addressing the confusions surrounding Table 1 and Figure 5.

---

### Decision · Program_Chairs · 2023-10-07

**Decision:**

Accept-Findings

**Comment:**

This paper presents a method to detect and analyze news media storms, which are periods of intense and focused coverage of a certain event, issue, or topic. The paper uses a large corpus of news articles from various sources and a neural network model to measure the similarity between articles. The paper then clusters the articles based on their similarity, time, and named entities to identify the media storms. The paper also examines the characteristics and dynamics of the media storms, such as their topics, duration, diversity, and inter-media influence. Overall, the paper makes contribution towards proposing a scalable news similarity detection model and the availability of the resulting dataset can help subsequent future works on media storms.

I thank the reviewers for providing useful suggestions and the authors for providing relevant responses to the concerns raised. I encourage the authors to incorporate these discussions in the future versions of the paper, such as, providing the rationale for using particular portions of the documents for similarity computation or for the specific choice of parameters for determining media storm, apart from addressing the confusions surrounding Table 1 and Figure 5.